# Quantitative Diffusion-Weighted Imaging Analyses to Predict Response to Neoadjuvant Immunotherapy in Patients with Locally Advanced Head and Neck Carcinoma

**DOI:** 10.3390/cancers14246235

**Published:** 2022-12-17

**Authors:** Hedda J. van der Hulst, Joris L. Vos, Renaud Tissier, Laura A. Smit, Roland M. Martens, Regina G. H. Beets-Tan, Michiel W. M. van den Brekel, Charlotte L. Zuur, Jonas A. Castelijns

**Affiliations:** 1Department of Radiology, The Netherlands Cancer Institute, 1066 CX Amsterdam, The Netherlands; 2GROW School for Oncology and Developmental Biology, University of Maastricht, 6211 LK Maastricht, The Netherlands; 3Department of Head and Neck Surgery, The Netherlands Cancer Institute, 1066 CX Amsterdam, The Netherlands; 4Immunogenomics and Precision Oncology Platform, Memorial Sloan Kettering Cancer Center, New York, NY 10065, USA; 5Human Oncology and Pathology Program, Memorial Sloan Kettering Cancer Center, New York, NY 10065, USA; 6Biostatistics Unit, The Netherlands Cancer Institute, 1066 CX Amsterdam, The Netherlands; 7Department of pathology, The Netherlands Cancer Institute, 1066 CX Amsterdam, The Netherlands; 8Department of Radiology and Nuclear Medicine, Amsterdam University Medical Center (AUMC), Location VUmc, 081 HV Amsterdam, The Netherlands; 9Department of Regional Health Research, University of Southern Denmark, 5230 Odense, Denmark; 10Department of Oral and Maxillofacial Surgery, Amsterdam University Medical Center (AUMC), Location AMC, 1105 AZ Amsterdam, The Netherlands; 11Department of Otorhinolaryngology Head Neck Surgery, Leiden University Medical Center, 2333 ZA Leiden, The Netherlands; 12Division of Tumor Biology and Immunology, The Netherlands Cancer Institute, 1066 CX Amsterdam, The Netherlands

**Keywords:** squamous cell carcinoma of head and neck, magnetic resonance imaging, diffusion magnetic resonance imaging, immune checkpoint blockade, immunotherapy, radiomics

## Abstract

**Simple Summary:**

Immunotherapy may induce early treatment response in head and neck squamous cell carcinoma (HNSCC) for some patients. Routine imaging parameters fail to diagnose these responses; however, magnetic resonance (MR) diffusion-weighted imaging (DWI) may be able to do so. This study sought to correlate DWI parameters with treatment response early after immunotherapy treatment in HNSCC. We analyzed 24 patients with advanced HNSCC with imaging before and after the immunotherapy. We found that rounder tumors that were smaller in diameter before treatment were more likely to respond. A decrease in skewness of the tumor after treatment compared to before treatment, as well as an overall low skewness post-treatment, were linked to better treatment response. Though this study was explorative in nature, these results are promising for the predictive use of MR-DWI in HNSCC treated with immunotherapy.

**Abstract:**

Background: Neoadjuvant immune checkpoint blockade (ICB) prior to surgery may induce early pathological responses in head and neck squamous cell carcinoma (HNSCC) patients. Routine imaging parameters fail to diagnose these responses early on. Magnetic resonance (MR) diffusion-weighted imaging (DWI) has proven to be useful for detecting HNSCC tumor mass after (chemo)radiation therapy. METHODS: 32 patients with stage II–IV, resectable HNSCC, treated at a phase Ib/IIa IMCISION trial (NCT03003637), were retrospectively analyzed using MR-imaging before and after two doses of single agent nivolumab (anti-PD-1) (*n* = 6) or nivolumab with ipilimumab (anti-CTLA-4) ICB (*n* = 26). The primary tumors were delineated pre- and post-treatment. A total of 32 features were derived from the delineation and correlated with the tumor regression percentage in the surgical specimen. Results: MR-DWI data was available for 24 of 32 patients. Smaller baseline tumor diameter (*p* = 0.01−0.04) and higher sphericity (*p* = 0.03) were predictive of having a good pathological response to ICB. Post-treatment skewness and the change in skewness between MRIs were negatively correlated with the tumor’s regression (*p* = 0.04, *p* = 0.02). Conclusion: Pre-treatment DWI tumor diameter and sphericity may be quantitative biomarkers for the prediction of an early pathological response to ICB. Furthermore, our data indicate that ADC skewness could be a marker for individual response evaluation.

## 1. Introduction

Head and neck squamous cell carcinoma (HNSCC) accounts for approximately 4% of worldwide cancer cases. Known risk factors for HNSCC development are the consumption of tobacco and alcohol, and infection with human papillomavirus (HPV) [1] Current curative treatment options for HNSCC patients are definitive radiotherapy with or without cisplatin-based chemotherapy [(C)RT], or primary surgery with or without adjuvant (C)RT. Despite these invasive treatments, advanced stage HNSCC patients have a poor prognosis [2,3]: their 5-year overall survival (OS) after extensive (salvage) surgery and adjuvant (C)RT is only 40–53%, thus underpinning the yet-unmet need for other treatment options [4]. 

Immune checkpoint blockade (ICB) of programmed cell death protein 1 (PD-1) therapy nearly triples the 2-year overall survival rate compared to investigator’s choice in platinum-refractory, recurrent, or metastatic HNSCC patients [5,6]. In a curative setting, pre-surgery, neoadjuvant anti-PD-1 monotherapy alone or combined with cytotoxic T-lymphocyte-associated protein 4 (CTLA-4) ICB induces a major response, with >90% pathological tumor regression in 20–35% of HNSCC patients [7,8]. However, biomarkers predicting a response to neoadjuvant ICB are not available. Within the IMCISION trial, patients who developed a major pathological response (MPR) after neoadjuvant anti-PD-1 monotherapy or concurrent anti-PD-1 and anti-CTLA-4 all remained free of HNSCC at a median follow-up of two years [8], raising the question of whether extensive and potentially mutilating surgery is necessary after a major response to ICB. De-escalating or delaying surgery in this population, however, requires an accurate and, preferably, minimally invasive method to establish ICB response reliably. Evaluation by computed tomography (CT) and magnetic resonance (MR) imaging, according to the current response evaluation criteria in solid tumors (RECIST 1.1 [9]), unfortunately underestimates the frequency and depth of pathological response after neoadjuvant ICB in HNSCC [7,8,10,11], highlighting the need for other modalities to establish response early upon neoadjuvant treatment. 

In IMCISION, 18[F]-fluorodeoxyglucose (FDG)-positron emission tomography (PET)-based metabolic response evaluation has shown promise in detecting pathological responses in HNSCC patients treated with neoadjuvant ICB, using the delta of the calculated total lesion glycolysis (TLG) [12]. Although the accuracy at the primary tumor site is 94%, this technique is limited due to the relatively large tumor volume needed to assess TLG and FDG avidity for the influx of immune cells upon ICB treatment, which can lead to false positive results [13]. 

MR-imaging is a fundamental modality in clinical practice, providing extensive anatomical and functional information regarding soft tissue [14]. Even so, anatomic MR-imaging performs poorly in distinguishing post-treatment effects from tumor recurrence after (C)RT [13]. Diffusion-weighted imaging (DWI) is a functional MR technique able to assess random-water (Brownian) motion, which is predominantly hindered by cellularity and, thus, negatively correlated with highly cellular tumor tissue [15,16] DWI has shown value in differentiating between tumor recurrence or benign, post-(chemo)radiotherapy effects in HNSCC [13,17]. Studies have shown the value of pre-treatment functional MR imaging in predicting treatment outcome or HNSCC after (C)RT [18,19]. 

However, whether DWI can differentiate HNSCC tumor cells from the influx of immune cells in the context of a major response to neoadjuvant ICB remains unknown. In this study, multiple quantitative DWI radiomic parameters were assessed, and their ability to predict pre-treatment or to diagnose early post-treatment of a major pathological response to neoadjuvant ICB was explored in HNSCC patients. 

## 2. Materials and Methods

### 2.1. Patient Population and Trail Treatment Details

This retrospective study used the data from the 32 patients treated within the non-randomized, open-label phase Ib/IIa IMCISION trial carried out at the Netherlands Cancer Institute (NKI) between February 2017 and October 2019 (NCT03003637). The IMCISION trial included adult patients with primary or recurrent, advanced stage (T2–T4, N0–N3b, M0) HNSCC of the oral cavity, oropharynx, hypopharynx, or larynx, eligible for curative (salvage) surgery. Staging was reported according to the 8th edition of the American Joint Committee on Cancer (AJCC) staging manual. All patients had a World Health Organization (WHO) Performance Score (PS) of 0 or 1. Critical exclusion criteria were the presence of distant metastases, a medical history of autoimmune disease, immunodeficiency, hepatitis B or C virus infection, or the use of immunosuppressive medication prior to treatment. Both HPV-negative and HPV-positive HNSCC patients were allowed to enter the trial. Complete in- and exclusion criteria, methods, and main results if the IMCISION trial have previously been published [8]. 

As part of the phase Ib trail safety run-in, the first 6 patients were treated with nivolumab monotherapy (240 mg) in weeks 1 and 3 (NIVO MONO), followed by surgery in week 5–6. Upon establishing feasibility, defined as the absence of surgical delay due to immune-related toxicity beyond week 6, the following 6-phase Ib patients were treated with a combination of nivolumab (240 mg) and ipilimumab (1 mg/kg) in week 1 and nivolumab (240 mg) in week 3 (COMBO), prior to surgery in week 5–6. When the COMBO regimen proved feasible, phase IIa opened and included 20 extra patients treated with the COMBO regimen prior to surgery in week 5–6. If indicated according to treatment guidelines, adjuvant (C)RT was performed. The institutional review board of the NKI reviewed and approved the IMCISION trial. All patients provided written informed consent prior to enrolment in IMCISION, which covered the use and presentation of the patient data enclosed in this manuscript [8].

### 2.2. Mr Imaging Acquisition

The MR-examinations were acquired on a 1.5 or 3.0T MRI scanner (Philips Healthcare) at baseline (week 0, 17 days (±10) pre-treatment) and after neoadjuvant ICB (post-treatment) at the end of week 4, 3 days (±0.6) before surgery, using a 16-channel head and neck coil combined with neck attachment. In 3 patients, the baseline MR imaging obtained in the referring institution was used. An overview of the varying examination parameters can be found in Appendix A. 

The MR imaging protocol included 2D T1-weighted imaging (T1W) prior to contrast administration, a 2D T1W after contrast injection (T1Wc), and a diffusion-weighted imaging (DWI) sequence. Depending on the center and protocol, either a T2-weighted sequence (T2W), a T2W with short tau inversion recovery (STIR) image, or a T2W Spectral Attenuated Inversion Recovery (SPAIR) was included.

The T1W and T1Wc images were generated using a turbo spin echo (TSE), and had a slice thickness of 3 to 4 mm, an echo time (TE) of 10 ms, and a repetition time (TR) of 538 to 778 ms. Contrast was administered with a standard dose of 15 mL gadolinium (0.5 mM Dotarem, Guerbet, France), followed by a 20 mL saline flush.

DWI was acquired with b-values of either 0, 200, and 1000 s/mm^2^, or b-values of 0, 200, 400, 600, 800 s/mm^2^, depending on the protocol. The DWI had a slice thickness of 3 to 4 mm, a TE range between 67 and 80 ms, and a TR range between 3658 and 5255 ms. The apparent diffusion coefficient (ADC) map was automatically calculated on a pixel-by-pixel basis by the Philips Healthcare-provided MRI software using 3 b-values: 0, 200, and either 800 or 1000 s/mm^2^, depending on availability (Appendix A).

### 2.3. Postprocessing and Feature Extraction

All available MR-DWI scans were delineated by physician researcher (H.v.d.H.) under supervision of an experienced H&N radiologist (J.C.). Regions of interest (ROIs) were manually placed on the entire tumor volume using 3DSlicer software (Version 4.10.2; http://www.slicer.org [20]). ROIs were placed on the solid tumoral components according to high signal intensity on DW MR-images acquired by the highest available b-value (either 800 or 1000 s/mm^2^), combined with corresponding low intensity on the ADC map [13]. Window and level values were kept consistent between patients. Large cystic or necrotic areas were excluded in order to focus on the viable tumor cells. T1Wc and STIR or SPAIR were used for anatomical correlation and tumor location. Researchers were blinded for treatment outcome. In Figure 1, an example of a delineated tumor is shown on DWI and ADC maps.

Signal intensity normalization was not needed for calculation of ADC map data, as this had already been derived from the DWI data. Image resampling was set to isotropic voxels of 2.0  mm. Bin width for the quantification of texture images was set to 5. Due to the variability in examination parameters and the small sample size, only first order and shape features were extracted using open-source package PyRadiomics 2.2.0 (Amsterdam, the Netherlands) [21]. Thus, a total of 32 radiomic features per patient was calculated from the ADC map data.

### 2.4. Outcome Assessment

The primary outcome of pathological response to neoadjuvant immunotherapy was assessed by an experienced head and neck pathologist (L.S.), using histological examination of the H&E-stained surgically resected specimen versus baseline biopsy specimen [8].

In the baseline biopsies and resection specimens, the percentage of viable tumor cells was calculated by measuring the area of viable tumor cells divided by the entire tumor bed area (defined by necrosis, fibrosis, keratinous debris, scarring, and immunoreaction sites). Since previously (C)RT-treated, salvaged patients had already low viable tumor cell count at baseline biopsies, the change in viable tumor cell percentage from baseline biopsy to the post-treatment resection specimen was calculated in all patients, and, henceforth, was called the tumor regression percentage [22].

Patients with ≤10% residual viable tumor cells and 90–100% of tumor regression percentage were defined as major pathological responders (MPRs). Patients with ≤50% residual viable tumor cells and 50–89% regression percentage had a partial pathological response (PPR), and patients with any percentage of residual viable tumor cells, but <50% tumor regression percentage, had no pathological response (NPR) [23].

In contrast to patients with a PPR and NPR, patients with an MPR after neoadjuvant immunotherapy in IMCISION were characterized by an excellent clinical outcome, without recurrent disease, at a median follow-up of 2 years [8]. Consequently, patients with a primary tumor MPR were considered responders, whereas patients with PPR or NPR were considered non-responders. Two patients were not available for pathological evaluation as they did not undergo curative surgery [8]. These patients were classified according to their response as defined by the radiological RECIST-criteria on the post-treatment MRI, 10 to 16 days after the last ICB infusion, combined with clinical follow-up. 

### 2.5. Statistical Analysis

Analyses were conducted using R (R Core Team (2022), R: A language and environment for statistical computing, R Foundation for Statistical Computing, Vienna, Austria, https://www.R-project.org/, version 4.1.2). As the radiomics features are extracted from MR-imaging of different scanners, the stability of the radiomic features between the two predominately used MR-machines was assessed using the one-way ANOVA test. A *p*-value < 0.05 was considered significant for the instability of the feature. 

Because of the small sample (<30 patients), and the fact that the data are not normally distributed, the means could not be compared using a Student’s t-test. A Wilcoxon signed rank test would not provide effect sizes for the data. Regression methods were used to analyze potential associations between the radiomics features and the pathological response. 

Regression analyses were applied for two classifications of pathological outcome; the continuous rate of pathological tumor regression (0–100%) was analyzed using linear regression, and the two defined responders (MPR) and non-responders groups (NPR + PPR) were analyzed using logistic regression. Considering the small number of patients and most clinical variables being of the categorical type, only 2 covariates (age and sex) were included in the multivariate analyses.

A separate regression analysis was performed separately for each feature at the baseline and post-treatment time points, and for the delta of each feature between the time points. The delta features were calculated only when both baseline and post-treatment data were available per patient. The significance level for the regression analyses was set at *p* < 0.05. Due to the explorative nature of the study, we chose not to correct for multiple testing.

Finally, we explored the predictive value of the pathological regression groups based on the evolution of all radiomics features together. To do so, logistic regression was used, and regularization penalization was optimized using elastic net. To minimize overfitting, cross-validation was applied by splitting the dataset on several folds. Considering the small sample size, a 3-fold cross-validation was used. The performance of the model for detecting response was assessed for each receiver operating characteristic (ROC) and area under the ROC curve (AUC), combined with specificity and sensitivity values.

## 3. Results

Two of the thirty-two patients enrolled in IMCISION had no DW MR imaging available, and were excluded. Six patients were excluded due to large artefacts or poor DWI quality at the tumor location on pre- and post-treatment imaging. Thus, twenty-four eligible patients with pre-treatment, post-treatment, or both time points’ DWIs remained for analysis (total of 44 MRI-examinations) (Figure 2). The mean age was 62 (±12) years, and 67% (*n* = 16) were male. All patients had oral cavity or oropharyngeal tumors. One patient had an HPV-positive HNSCC; all other tumors were HPV-negative. Seven patients were included with recurrent or residual HNSCC after previous (C)RT (*n* = 3) or previous surgical treatment, with or without adjuvant RT (*n* = 4). On average, patients were diagnosed with a T3 tumor (45.8%), N0 stage (54.2%), and with disease stage III-IV (66.7%). Of the 24 patients, 7 had MPR to treatment, 15 patients had NPR, and 2 patients had PPR. See Table 1 for all baseline characteristics. 

From the ADC-map data, 32 radiomic features were extracted. After analyzing the stability of the data throughout the different scanners, 10 features were deemed unstable (*p* < 0.05) and were excluded from further analyses. (Appendix B) The remaining 22 stable features were analyzed for possible associations with the outcome. The mean values of these remaining features at the different time points are shown in Table 2. 

Based on baseline imaging, responding patients had significantly smaller tumor diameters prior to therapy, measured from 3D and 2D dimensions on varying planes, compared to patients who did not respond (*p* = 0.04). Furthermore, these patients had an overall more spherical (round) tumors (*p* = 0.03) and lower entropy values (*p* = 0.05) than non-responders (Figure 3, Table 3). Entropy specifies the randomness in the image values, where lower entropy means more homogeneous tissue [21]. These variables, except for tumor sphericity, were likewise associated with the continuous tumor pathological regression percentage (Table 4). All findings were unaffected after correcting for sex and age (Table 3 and Table 4).

Upon post-treatment imaging after immunotherapy and prior to surgery, sphericity and entropy were no longer correlated with response. The post-treatment analysis did yield significant difference in three of the diameter measurements: the 3D diameter (*p* = 0.05), the diameter in the sagittal plane (row) (*p* = 0.04), and the major axis length (*p* = 0.04) between responders and non-responders (Table 3). Similar results were seen for the tumor pathological regression percentage. In addition, a new, significant difference was seen for skewness of the ADC, as negative skewness was associated with higher pathological response percentage in the univariate analyses (*p* = 0.04) as well as in the multivariate analyses (*p* = 0.05).(Table 4)

In the multivariate analysis of the delta between the features collected from pre- and post-treatment DWI, ADC skewness was the only significant feature that was correlated to tumor regression percentage upon ICB (*p* = 0.02) (Table 4). However, skewness was not correlated to either of the two response groups (*p* = 0.07) (Table 3).

As the degree of pathological tumor regression is based on changes in viable tumor cells in the tumor bed between the baseline biopsy and the surgical specimen, we would expect the delta of the features post-treatment to depict this regression most accurately. 

However, except for ADC skewness, no significant associations between the delta features were observed. Nonetheless, combining all delta features together in a model did yield a considerable AUC of 0.846 (0.716–0.977) in predicting responders from all non-responders (Appendix C).

## 4. Discussion

This study aimed to explore the ability of MR diffusion-based imaging parameters, extracted from MR-imaging acquired at baseline and after immunotherapy treatment (shortly prior to surgery), to predict or detect major pathological responses to neoadjuvant ICB early on in patients with resectable HNSCC.

Several differences were seen at the baseline between the pathological response groups in our data. Responding tumors were characterized by a significantly smaller diameter, greater sphericity, and lower entropy values. 

Higher sphericity (tumor roundness) has previously been correlated to improved progression-free survival (PFS) and local control after (C)RT in HNSCC [24,25,26]. Sphericity is associated with an expansive pattern of tumoral growth often seen in HPV-positive HNSCC, in contrast to the infiltrative growth pattern more common in HPV-negative tumors [24,27,28]. As baseline DWI was unavailable for the only HPV-positive patient in this dataset, the higher sphericity observed in the ICB-responsive population was fully reflective of HPV-negative HNSCC. In this cohort, necrotic area(s) were excluded in the delineation, creating a more complex and, thus, less spherical shape in necrotic tumors. This is illustrated in Figure 4a,b. This may result in possible confounding of the tumor sphericity, as this feature may now also be negatively associated with the baseline presence of necrotic area(s) within the tumor. As poor perfusion may impede the delivery of intravenous drugs [29], necrotic (non-spherical) tumors may have been overrepresented in the non-responding group. In addition, the inclusion of patients with recurrent or residual disease after prior RT may also have influenced the sphericity, due to previous treatment effects on the tumoral area resulting in regional necrosis or fibrosis, though our data showed no clear trend for sphericity. 

On the contrary, the highest entropy values of all evaluated tumors were discernable for the previously treated cancers. Entropy is a statistical measure of the randomness in the image values used to characterize the texture of the input image [21]. A lower level of entropy is defined by lower levels of chaos and randomness, and resembles more homogenous tissue, whereas higher entropy is linked to heterogeneity of the tissue. Though heterogeneous tumors have also been linked to higher chances of local failure, our data may have been confounded by the inclusion of previously irradiated tumors [25,30]. Finally, the diameter parameter, measured in different planes and dimensions, was significantly different between the two response groups. Correlations between pre-treatment smaller tumor volume and increased locoregional control or PFS have often been described for (C)RT treated patients [31,32,33,34]. Initially, it was hypothesized that a significant change in volume would occur based on the applied delineation method of low ADC combined with high b1000/800 areas, which should result in a decline in tumor size at post-treatment as a result of a decrease in tumor cellularity density upon response [13]. The maximum diameter can even be fairly reliably used for tumor volume estimation in anatomical MRIs [35]. Yet, in contrast to the diameter, no significance was seen for tumor volume throughout our dataset, though a lower baseline *p*-value (*p* = 0.1) was discernable using the tumor regression percentage (Table 4). When considering the substantial standard deviation of the volume parameter, as shown in Table 2, it becomes apparent that the range of the volume parameter is too broad for this small dataset to be significant, though an overall smaller tumor volume can be seen in responders compared to non-responders. The diameter appears to be less variable between patients, and is, therefore, significant within this cohort. 

At post-treatment imaging, most significant correlations dissipated, with the exception of the newly found parameter skewness of the ADC and the maximum diameter in 3D, the sagittal plane and major axis. Intriguingly, a recent study describes a moderate inverse correlation between ADC skewness and programmed death-ligand 1 (PD-L1) expression scores [36] Though this correlation alone is not strong enough to predict PD-L1 expression in a clinical routine, it does indicate the ADC-map may depict more complex histopathology than just the viable tumor cells. Moreover, this correlation is especially interesting since nivolumab functions as a PD-1 inhibitor. A significantly lower post-treatment skewness or a decrease in the skewness during treatment, as seen with the delta analyses, may have some usefulness in treatment monitoring in the future [8]. However, within this dataset, significance was only observed for the continuous tumor regression percentage, and not as clearly in the two defined response groups. The maximum diameter, measured in several dimensions and planes, was also significant post-treatment, but less so than at baseline. This could be due the challenges of delineating post-treatment, as is shown by increased variation between delineation post-treatment, and as illustrated in Figure 1 and Figure 4.

We constructed a model that yielded a high AUC, underlining the potential value of these parameters when assessed in concert, rather than individually. However, this model was trained and tested on the same limited dataset, and, therefore, was likely overfitted. Without external validation, its value remains uncertain. Recent work by Corino et al., however, promisingly illustrated a higher AUC of a CT radiomics model compared to a clinical model for predicting 10-month OS to nivolumab treatment of HNSSC-patients [37]. This highlights the potential for models based on MR-radiomics that could additionally provide extensive data on soft tumoral tissue.

This retrospective analysis had several limitations, such as the small sample size, making the study explorative in nature. Within this limited dataset, previously (C)RT-treated tumors were also included. These are radiologically different from primary tumors, which may have diluted the results. Furthermore, the MR-imaging was not all acquired according to the same protocol, on the same MR-machine, or in the same treatment center. While stability between the two main MRI types and field strength was established, and instable features excluded, minor variations may still have limited the interpretability of the results. In addition, some patients were excluded based on artefacts, indicating that a possible MRI model will not be applicable for everyone. No correlation between molecular pathological biomarkers and imaging markers was explored due to the limited sample size. 

Neoadjuvant immunotherapy trials employing combined aPD1 and aCTLA4 immune-checkpoint blockade prior to extensive curative surgery and radiation therapy provided promising results, namely a MPR rate of 30% upon immunotherapy at the time of surgery, at the primary tumor site [7,8]. This study examined a novel cohort of immunotherapy-treated HNSCC, and attempted to explore the possible value of DWI for response monitoring after immunotherapy in HNSCC. In view of future clinical trials aiming at de-escalation of surgery in these patients, biomarkers to identify these patients with a favorable response upon immunotherapy prior to surgery are needed. RECIST 1.1 [9] criteria used for (chemo) radiation-type treatments do not currently suffice to monitor treatment response in immunotherapy, suggesting a different imaging approach for immunotherapy response monitoring [7,8,10,11]. Though more research in larger datasets is required to definitively link DWI parameters to immunotherapy outcomes, this study might provide guidance for the direction of such research. 

## 5. Conclusions

In conclusion, while this study has certain limitations, our analysis of DWI features identified a significant association between baseline tumor diameter and sphericity and pathological response after neoadjuvant nivolumab and ipilimumab in HNSCC. In addition, ADC skewness may play an important role in pathological response evaluation, either as a stand-alone post-treatment value or calculated as the difference between baseline and post-treatment imaging.

## Figures and Tables

**Figure 1 cancers-14-06235-f001:**
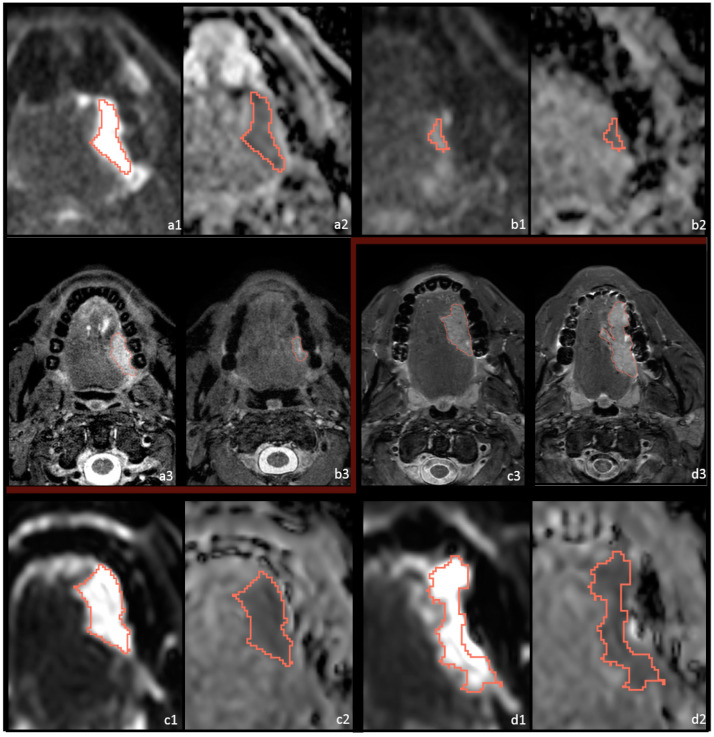
DWI and STIR imaging of 2 patients before and after immunotherapy treatment with different treatment outcomes. Pt39, with a primary cT3N0 HNSCC of the oral cavity, is depicted before ((**a1**) b1000; (**a2**) ADC-map; (**a3**) STIR) and after ((**b1**) b1000; (**b2**) ADC-map; (**b3**) S = TIR) immunotherapy; this patient had MPR to treatment. Pt37 with a primary cT3N0 HNSCC of the oral cavity is depicted before ((**c1**) b1000; (**c2**) ADC-map; (**c3**) STIR) and after ((**d1**) b1000; (**d2**) ADC-map; (**d3**) STIR) immunotherapy; this patient had NPR to treatment.

**Figure 2 cancers-14-06235-f002:**
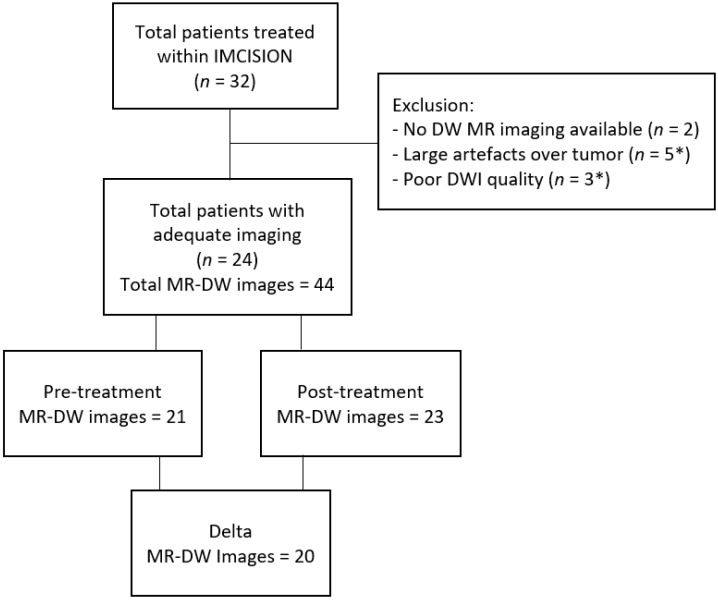
Flow diagram of the patient inclusion. MR-DW: magnetic resonance diffusion weighted, *: Some patients had both a large artefact and low DWI quality.

**Figure 3 cancers-14-06235-f003:**
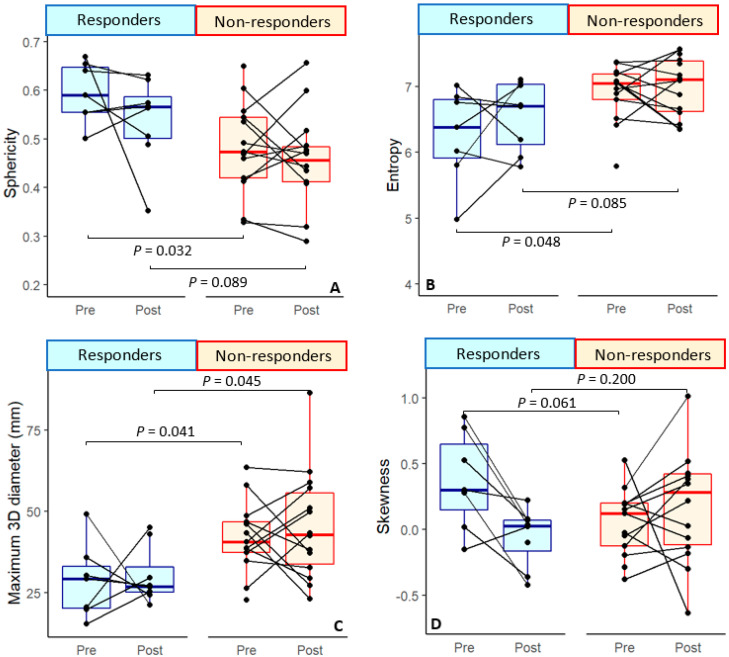
Boxplot overview of some correlations between responder and non-responder groups. (**A**) Sphericity or tumor roundness is significantly higher for responders at pre-treatment; this dissipates at post-treatment imaging. (**B**) Significantly lower levels of entropy can be seen for responders compared to non-responders, but only at pre-treatment imaging. (**C**) The maximum diameter measured using 3D dimensions is significantly lower at pre-treatment imaging for responders compared to non-responders. (**D**) No significance of ADC skewness can be seen between the two response groups at pre- or post-treatment. A near-significant result (*p* = 0.066) is observed for the delta of ADC skewness.

**Figure 4 cancers-14-06235-f004:**
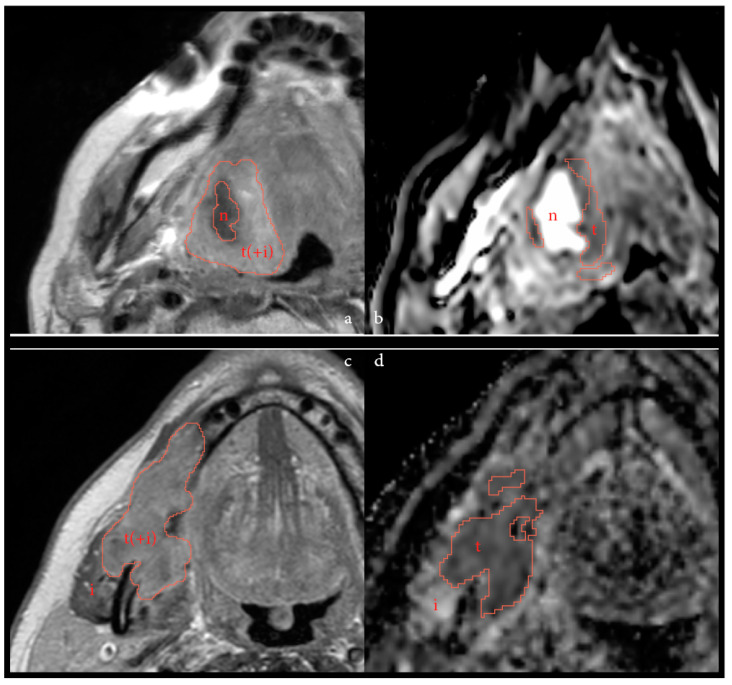
Two different patients illustrating possible tumor microenvironments on T1Wc imaging and on ADC. Figure 4a and b depict pre-immunotherapy imaging of a patient with a residual tumor (rT3N0) after previous CRT with a necrotic area (*n*) within the tumor region (t) on T1Wc (**a**), and as shown on ADC-map (**b**). On the T1Wc imaging, a possible infiltrate (i) could be included in the tumor area, marked by t(+i), as this is difficult to discern. This patient did not respond to treatment. Figure 4c and d depict post-treatment imaging of a cT4N1 primary tumor of the oral cavity, with possible immune infiltrate (i) surrounding the tumor (t), or, conceivably, within the tumor area (t(+i)) on T1wc (**c**) and ADC-map (**d**). This patient had a partial pathological response to treatment.

**Table 1 cancers-14-06235-t001:** HPV: human papillomavirus, OSCC: oropharyngeal squamous cell carcinoma, *n*: sample size, *n*: subsample size, AJCC: American Joint Committee on Cancer. NIVO MONO: nivolumab monotherapy, COMBO: combined nivolumab and ipilimumab.

Baseline Characteristics (*n* = 24 Patients)				
		All (*n* = 24)	Responders (*n* = 8)	Non-Responders (*n* = 16)
Age (years)	Mean (SD)	62.3 ± 12.1	61.6 ± 8.6	61.3± 13.9
Sex	Male	16 (67%)	6 (75%)	10 (62.5%)
	Female	8 (33%)	2 (25%)	6 (37.5%)
HPV-status	Positive	1 (4%)	1 (12.5%)	0
	Negative	23 (96%)	7 (87.5%)	16 (100%)
Smoking	Never	4 (17%)	1 (12.5%)	3 (18.8%)
	Currently	8 (33%)	2 (25%)	6 (37.5%)
	Quit >2 years ago	12 (50%)	5 (62.5%)	7 (43.8%)
Tumor location	Oral cavity	21 (88%)	7 (87.5%)	14 (87.5%)
	Oropharynx	3 (13%)	1 (12.5%)	2 (12.5%)
Tumor status	Primary	17 (71%)	7 (87.5%)	10 (62.5%)
	Recurrent	4 (17%)	1 (12.5%)	3 (18.8%)
	Residual	3 (13%)	0	3 (18.8%)
Clinical T-stage	T2	4 (17%)	2 (25%)	2 (12.5%)
	T3	11 (46%)	4 (50%)	7 (43.8%)
	T4a	9 (38%)	2 (25%)	7 (43.8%)
Clinical *n*-stage	N0	13 (54%)	4 (50%)	9 (56.3%)
	N1	6 (25%)	3 (50%)	2 (12.5%
	N2a	1 (4%)	0	1 (6.3%)
	N2b	3 (13%)	0	3 (18.8%)
	N2c	1 (4%)	0	1 (6.3%)
AJCC disease stage	II	1 (4%)	1 (12.5%)	0
	III	8 (33%)	5 (62.5%)	3 (18.8%)
	IV	8 (33%)	1 (12.5%)	7 (43.8%)
	Recurrent	7 (29%)	1 (12.5%)	6 (37.5%)
Immunotherapy regimen	NIVO MONO	5 (21%)	1 (12.5%)	4 (25%)
	COMBO	19 (79%)	7 (87.5%)	12 (75%)
Surgical treatment	Yes	22 (92%)	7 (87.5%)	15 (93.8%)
	No	2 (8%)	1 (12.5%)	1 (6.2%)

**Table 2 cancers-14-06235-t002:** Mean value of responding and non-responding tumor features at pre- and post-treatment timing, and the calculated delta features using the patients with data available at both time points. Pre-tr: pre-treatment, On-tr: post-treatment. *: Significant in analyses in Table 3.

Mean of Responding and Non-Responding Tumor Features at Pre- and Post-Treatment and the Calculated Delta						
	Responders			Non Responders		
	Pre-tr	Delta	Post-tr	Pre-tr	Delta	Post-tr
	(*n* = 7)	(*n* = 7)	(*n* = 8)	(*n* = 14)	(*n* = 13)	(*n* = 15)
**First-order Parameters**	Mean ± SD	Mean ± SD	Mean ± SD	Mean ± SD	Mean ± SD	Mean ± SD
Min ADC (×10^−3^ mm^2^/s)	0.49 ± 0.28	0.01 ± 0.35	0.44 ± 0.28	0.42 ± 0.21	−0.07 ± 0.25	0.29 ± 0.22
10th percentile	0.75 ± 0.24	0.03 ± 0.33	0.75 ± 0.23	0.81 ± 0.14	−0.05 ± 0.13	0.74 ± 0.13
Energy (×10^8^)	6.62 ± 8.33	−2.64 ± 9.57	5.02 ± 3.74	11.78 ± 8.81	3.67 ± 12.70	26.45 ± 48.32
Total energy (×10^8^)	52.94 ± 66.67	−21.08 ± 76.54	40.12 ± 29.89	94.21 ± 70.50	29.32 ± 101.58	211.6 ± 386.6
Entropy	6.26 ± 0.71 *	0.22 ± 0.74	6.56 ± 0.53	6.89 ± 0.41 *	−0.06 ± 0.45	6.99 ± 0.44
Skewness	0.37 ± 0.37	−0.42 ± 0.36	−0.06 ± 0.22	0.08 ± 0.25	0.02 ± 0.47	0.15 ± 0.41
Kurtosis	3.49 ± 0.84	−0.64 ± 0.97	2.99 ± 0.52	3.36 ± 0.63	0.38 ± 1.05	3.72 ± 0.95
Uniformity	0.016 ± 0.008	−0.003 ± 0.008	0.013 ± 0.005	0.010 ± 0.003	0.001 ± 0.003	0.010 ± 0.003
**Shape Parameters**						
Volume (cm^3^)	2.81 ± 3.46	−1.39 ± 3.85	2.29 ± 2.55	4.24 ± 3.13	1.81 ± 5.69	10.97 ± 20.12
Voxel volume (cm^3^)	2.92 ± 3.51	−1.38 ± 3.92	2.43 ± 2.61	4.44 ± 3.21	1.83 ± 5.76	11.23 ± 20.21
Surface area (cm^2^)	14.87 ± 13.0	−3.37 ± 15.98	15.13 ± 11.59	26.81 ± 17.95	6.85 ± 25.79	46.63 ± 56.67
Surface area/volume ratio	0.75 ± 0.26	0.11 ± 0.36	0.81 ± 0.22	0.72 ± 0.21	0.02 ± 0.24	0.70 ± 0.26
Sphericity	0.60 ± 0.06 *	−0.05 ± 0.13	0.54 ± 0.09	0.49 ± 0.09 *	−0.002 ± 0.09	0.46 ± 0.09
3D diameter (cm)	2.86 ± 1.15 *	−0.01 ± 1.70	3.03 ± 0.88 *	4.15 ± 1.09 *	−0.08 ± 1.39	4.51 ± 1.64 *
2D diameter (Slice) (cm)	2.37 ± 1.01 *	0.02 ± 1.44	2.53 ± 0.71	3.47 ± 0.92 *	−0.003 ± 1.14	3.86 ± 1.62
2D diameter (Column) (cm)	2.30 ± 0.88	−0.25 ± 1.08	2.29 ± 0.90	2.86 ± 1.02	0.15 ± 0.99	3.43 ± 1.63
2D diameter (Row) (cm)	2.50 ± 1.12	−0.22 ± 1.37	2.52 ± 0.76 *	3.57 ± 1.04	−0.05 ± 1.37	3.96 ± 1.62 *
Major axis length (cm)	2.41 ± 0.84 *	0.16 ± 1.44	2.73 ± 0.81 *	3.47 ± 0.85 *	0.12 ± 1.32	3.85 ± 1.16 *
Minor axis length (cm)	1.82 ± 0.67	−0.06 ± 0.66	1.85 ± 0.50	2.23 ± 0.72	−0.05 ± 0.62	2.55 ± 1.18
Least axis length (cm)	0.94 ± 0.50	−0.05 ± 0.61	1.04 ± 0.51	1.49 ± 0.61	−0.06 ± 0.47	1.78 ± 1.09
Elongation	0.76 ± 0.16	−0.05 ± 0.23	0.70 ± 0.17	0.66 ± 0.21	−0.06 ± 0.14	0.65 ± 0.16
Flatness	0.38 ± 0.07	−0.02 ± 0.13	0.38 ± 0.14	0.43 ± 0.14	−0.01 ± 0.12	0.44 ± 0.16

**Table 3 cancers-14-06235-t003:** Features analyses of responder groups at pre-treatment, post-treatment, and delta time points.

Analyses of Features of Responding Tumors Versus Non-Responding Tumors (*n* = 24)										
	Pre-Treatment				Delta		Post-Treatment			
	Univariate		Multivariate †		Univariate		Univariate		Multivariate †	
**First-order Parameters**	*p*	β ± SE	*p*	β ± SE	*p*	β ± SE	*p*	β ± SE	*p*	β ± SE
Min ADC (10^−3^ mm^2^/s)	0.366				0.551		0.171			
10th percentile	0.520				0.416		0.869			
Energy (×10^8^)	0.282				0.268		0.195			
Total Energy (×10^8^)	0.282				0.268		0.195			
Entropy	0.048 *	−1.43 ± 0.69	0.033 *	−6.32 ± 2.97	0.285		0.067			
Skewness	0.061				0.066		0.200			
Kurtosis	0.712				0.061		0.062			
Uniformity	0.076				0.177		0.075			
**Shape Parameters**										
Volume (cm^3^)	0.343				0.212		0.197			
Voxel volume (cm^3^)	0.323				0.216		0.192			
Surface area (cm^2^)	0.162				0.347		0.136			
Surface area/volume ratio	0.769				0.493		0.321			
Sphericity	0.032 *	1.84 ± 0.86	0.024 *	2.57 ± 1.138	0.327		0.089			
3D diameter (cm)	0.041 *	−1.47 ± 0.72	0.034 *	−2.29 ± 1.08	0.913		0.045 *	−1.75 ± 0.87	0.040 *	−1.88 ± 0.91
2D diameter (Slice) (cm)	0.038 *	−1.40 ± 0.68	0.028 *	−2.16 ± 0.98	0.961		0.056			
2D diameter (Column) (cm)	0.226				0.398		0.104			
2D diameter (Row) (cm)	0.072				0.786		0.044 *	−2.34 ± 1.16	0.047 *	−2.440 ± 1.23
Major axis length (cm)	0.038*	−1.57 ± 0.76	0.035 *	−2.121 ± 1.01	0.949		0.044 *	−1.34 ± 0.67	0.042 *	−1.428 ± 0.71
Minor axis length (cm)	0.222				0.959		0.139			
Least axis length (cm)	0.079				0.630		0.114			
Elongation	0.304				0.848		0.450			
Flatness	0.324				0.914		0.350			

*: statistically significant, †: adjusted for age and sex. β: effect size regression coefficient, SE: standard error.

**Table 4 cancers-14-06235-t004:** Features analyses of the continuous (0–100) tumor regression percentage at pre-treatment, post-treatment, and delta time points.

Continuous (0–100) Tumor Regression Percentage Analyses with Feature (*n* = 22)												
	Pre-Treatment				Delta				Post-Treatment			
	Univariate		Multivariate †		Univariate		Multivariate†		Univariate		Multivariate †	
**First order Parameters**	*p*	β ± SE	*p*	β ± SE	*p*	β ± SE	*p*	β ± SE	*p*	β ± SE	*p*	β ± SE
Min ADC (10^−3^ mm^2^/s)	0.437				0.868				0.201			
10th percentile	0.569				0.340				0.565			
Energy (×10^8^)	0.081				0.644				0.264			
Total Energy (×10^8^)	0.081				0.644				0.264			
Entropy	0.046 *	−17.80 ± 9.93	0.024 *	−23.87 ± 9.53	0.240				0.085			
Skewness	0.061				0.016 *	−20.88 ± 7.73	0.024 *	−21.63 ± 8.52	0.037 *	−17.37 ± 7.75	0.048 *	−18.05 ± 8.49
Kurtosis	0.971				0.134				0.096			
Uniformity	0.060				0.141				0.075			
**Shape Parameters**												
Volume (cm^3^)	0.095				0.467				0.274			
Voxel volume (cm^3^)	0.090				0.964				0.271			
Surface area (cm^2^)	0.078				0.772				0.175			
Surface area/volume ratio	0.363				0.964				0.413			
Sphericity	0.076				0.396				0.162			
3D diameter (cm)	0.017 *	−21.44 ± 8.07	0.009 *	−26.47 ± 8.89	0.477				0.051			
2D diameter (Slice) (cm)	0.038 *	−19.18 ± 8.52	0.026 *	−23.39 ± 9.50	0.609				0.087			
2D diameter (Column)(cm)	0.219				0.629				0.083			
2D diameter (Row) (cm)	0.027 *	−20.59 ± 8.49	0.015 *	−26.37 ± 9.54	0.819				0.034 *	−18.94 ± 8.28	0.040 *	−19.43 ± 8.75
Major axis length (cm)	0.008 *	−23.55 ± 7.77	0.006 *	−26.85 ± 8.32	0.531				0.041 *	−18.84 ± 8.60	0.055	
Minor axis length (cm)	0.522				0.888				0.195			
Least axis length (cm)	0.007 *	−27.45 ± 9.00	0.004 *	−33.07 ± 9.8	0.528				0.113			
Elongation	0.051				0.327				0.580			
Flatness	0.214				0.923				0.287			

*: statistically significant, †: adjusted for age and sex. β: effect size regression coefficient, SE: standard error.

## Data Availability

The data presented in this study are available on request from the corresponding author.

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
