# Peer review of "Quantitative Diffusion-Weighted Imaging Analyses to Predict Response to Neoadjuvant Immunotherapy in Patients with Locally Advanced Head and Neck Carcinoma"

_cancers, 2022, doi:10.3390/cancers14246235_

Round 1

Reviewer 1 Report

The manuscript addresses a very important and relevant topic - the possibility to predict response to immune checkpoint inhibitors. The use of these antibodies in clinical practice is restricted by incomplete response rates and the possibility of adverse events (hyperprogression). More specifically, the question remains which of the tumors that respond may have a high risk of recurrence, and which may regress, and how soon? Since reliable markers are not available, non-invasive methods to monitor lesions post-therapy would be of immense value. 

I cannot find any fundamental flaws in the study design, especially not in the image analysis protocols used specifically for DWI images, and how the features were extracted and quantitated. At the same time, I am not an expert on MRI procedures and cannot comment on those. 

Different tumor features are mentioned, with differential imaging properties, such as necrosis, fibrosis, keratinous debris, scarring and immunoreaction sites. It should be considered to show examples for these different intra-tumor regions, and heterogeneity, in the manuscript; with a focus on the immune reaction sites. This could help the reader to appreciate the manuscript more, and intuitively understand the benefits and also the limitations of image analyses. Figure 1 shows a negative case, with no pathological response to the therapy; why not also show a positive case? 

The statistical method, based on logistic regression models, may need to be explained in more detail since the number of patients and especially the number of images available for analysis (n = 20, effectively; derived from 32 patients to begin with) is really small and the robustness of the statistical analysis is key. Also, 3-fold cross-validation may not be very robust (although there is not much of a choice here), but alternatives to the regression-based method should be considered and evaluated in comparison. This is also highlighted by the borderline statistical significance of the findings, e.g., shown in Fig. 3. These are just barely < 0.05 cutoff significance (or even above) and "don't look" very convincing. This is not unexpected, considering the small number of cases/patients and treatment protocols, but it requires a reproducible, more robust statistical evaluation including alternative approaches that clearly show why the current method was chosen. 

Table 2 is detailed and at the same time its a bit tedious to identify the significant associations across a sea of non-significant ones. If possible, this should be highlighted to make the reader's life easier. (Okay, I realize tables 3 and 4 provide that, but then the question arises if table 2 is really that informative an necessary?). 

Overall, I find the manuscript well written, and the data are described in a probably adequate way... it's just that these data are a bit thin. I can imagine that the continuation of such studies, to generate additional, more robust data, is limited or impossible, and may not be held against the authors. But under these conditions, I think some more transparency concerning the statistical evaluation is mandatory, with clear reasons why some methods are preferred over others. That way, its possible to recapitulate the process and publication even with restricted data seems possible. 

Such issues are actually discussed in the last paragraph, and rightly so.  I think, therefore, that the authors are fully aware of the limitations and they have nicely discussed these issues. Which is fair enough; it just makes the decision-making difficult: not only concerning the acceptance of this manuscript, but (of course) the overall benefit of the imaging procedures for the clinics. 

smallish things

The names of the trial is sometimes given as IMCISON and sometimes as IMCISION. Those are probably just typos? 

Reviewer 2 Report

The current study reported a statistical study to correlate quantitative Diffusion-Weighted Imaging (DWI) with the treatment response to neoadjuvant immunotherapy in patients with head and neck cancer (HNC). The data is solid and may provide some useful information in treatment of HNC. However, there are a few weaknesses in this study.

 (1)  As the author mentioned, the sample size is small.  It may need to use another cohort to confirm the conclusion.

(2)  It may need to discuss whether the conclusion obtained from this study is limited to immunotherapy or other chemo-neoadjuvant treatments since this study did not involve molecular pathological or biomarker evaluation.

(3)  There were grammar errors.  In page 6 line 233 and page 7 line 243, Arabic numerals cannot be used in the beginning of a sentence. 
